# Nanoparticle Effects on Ice Plant Mineral Accumulation under Different Lighting Conditions and Assessment of Hazard Quotients for Human Health

**DOI:** 10.3390/plants13050681

**Published:** 2024-02-28

**Authors:** Rūta Sutulienė, Aušra Brazaitytė, Martynas Urbutis, Simona Tučkutė, Pavelas Duchovskis

**Affiliations:** Institute of Horticulture, Lithuanian Research Centre for Agriculture and Forestry, Kauno 30, Kaunas Distr., 54333 Babtai, Lithuania; ausra.brazaityte@lammc.lt (A.B.); martynas.urbutis@lammc.lt (M.U.); simona.tuckute@lei.lt (S.T.); pavelas.duchovskis@lammc.lt (P.D.)

**Keywords:** copper oxide nanoparticle, Cu accumulation, foliar application, hazard quotients, HPS, ice plant, lighting, LED, mineral nutrients, NPs, risk assessment, zinc oxide nanoparticles, Zn accumulation

## Abstract

Nanotechnologies can improve plant growth, protect it from pathogens, and enrich it with bioactive and mineral substances. In order to fill the lack of knowledge about the combined environmental effects of lighting and nanoparticles (NPs) on plants, this study is designed to investigate how different HPS and LED lighting combined with CuO and ZnO NPs influence the elemental composition of ice plants (*Mesembryanthemum crystallinum* L.). Plants were grown in hydroponic systems with LED and HPS lighting at 250 ± 5 μmol m^−2^ s^−1^ intensity, sprayed with aqueous suspensions of CuO (40 nm, 30 ppm) and ZnO (35–45 nm, 800 ppm) NPs; their elemental composition was measured using an ICP–OES spectrometer and hazard quotients were calculated. LED lighting combined with the application of ZnO NPs significantly affected Zn accumulation in plant leaves. Cu accumulation was higher when plants were treated with CuO NPs and HPS illumination combined. The calculated hazard quotients showed that the limits are not exceeded when applying our selected concentrations and growth conditions on ice plants. In conclusion, ice plants had a more significant positive effect on the accumulation of macro- and microelements under LED lighting than HPS. NPs had the strongest effect on the increase in their respective microelements.

## 1. Introduction

Zinc (Zn) and copper (Cu) are crucial micronutrients in leafy vegetables and for human health. Zn plays a vital role in plant growth and development, and its presence in vegetables is influenced mainly by soil pH and fertilization [1,2]. Similarly, Cu is essential for enzymatic activities, cell metabolism, and signaling in plants, including leafy vegetables [3]. However, Zn and Cu deficiencies are prevalent in the human population, with more than half being deficient in Zn [4,5]. Therefore, it is crucial to ensure the leafy vegetables’ biofortification of these micronutrients to address these deficiencies.

Nanoparticles (NPs, materials with a size less than 100 nm) have the potential to address micro- and macronutrient scarcity by enhancing nutrient mobilization and uptake in plants [6,7]. Considering the differences between NPs and bulk materials, such as the larger surface area determined by the smaller size of NPs, the zeta potential, and the homogeneity of the suspension, these factors may impact their penetration and movement in plants. Engineered nanomaterials (ENMs) can be used as nanoherbicides and nanopesticides to detect agro-pathogens onsite, post-harvest management, soil fertility, and irrigation management [8]. Nanofertilizers, in particular, positively influence plant growth, development, and interactions with soil microflora [9]. Furthermore, ENMs can suppress plant disease and enhance crop yield, potentially due to the greater availability of nutrients in the “nano” form [10]. 

Singh [11] and Goyal [12] highlighted the potential of nano-enabled technology to enhance growth, nutritional quality, and Zn content in cereal crops and leafy vegetables. Sb [13] further demonstrated that 50 nm of ZnO NPs at a concentration of 1000 ppm increased spinach’s leaf length, width, leaf area, protein, and dietary fiber content. The researchers [14] determined the most appropriate concentrations of ZnO NPs application through the soil for lettuce—20 ppm, beet—225 ppm, wheat, bean, pea—450 ppm, maize, radish, tomato, and cucumber—900 ppm, plants based on the highest Zn content in the shoots of the studied plants. They highlighted that the effects varied depending on the plant species and especially on soil pH. However, Zhao [15] noted that applying 800 mg kg^−1^ of ZnO NPs led to the biggest Zn bioaccumulation in cucumber leaves, less in stems, and the least in fruits, suggesting a potential risk of excessive NP exposure. Dimkpa [16] demonstrated that the foliar application of ZnO NPs (18 nm, 6 ppm) more effectively increased Zn amount in sorghum shoots than application through the soil. Sharifan [17] reported that a 15–137 nm size in ZnO NPs at 100 ppm concentration significantly elevated the Zn contents in cilantro, parsley, and spinach. Ji [18] found that ZnO NPs—200 mg kg^−1^—and CuO NPs—25 mg kg^−1^—concentrations enhanced absorption of certain nutrients by 3 and 2.5 times, respectively, in *Medicago polymorpha* L compared with non-treated plants.

The effects of CuO NPs on leafy vegetables are complex and may vary depending on factors such as concentration and exposure duration. Pelegrino [19] found that 0.2–20 µg ml^−1^ concentrations of green synthesized CuO NPs (6.6 nm) can enhance lettuce growth, but 40–300 µg ml^−1^ concentrations can lead to decreased plant weight, net photosynthesis level, and water content, as well as the inhibition of seed germination and radicle growth. However, Wang [20] found that CuO NPs at 200 and 400 mg kg^−1^ increased Cu amount in shoots and roots, lettuce photosynthesis rate, and productivity. Zafar [21] noted that CuO NPs 53 nm in size and at 500 to 1500 ppm concentration inhibited root, stem, and leaf growth in *Brassica nigra* seedlings. Researchers [22] investigated that the highest amount of Cu was found after foliar application of CuO NPs (40–60 nm) at 250 ppm concentration exposure in lettuce and cabbage after 15 days. These findings collectively suggest that nanotechnology can be crucial in addressing Zn and Cu micronutrient scarcity in leafy vegetables. 

It is worth noting that most of the research has been conducted with various plants popular in horticulture, and NPs are usually applied through the soil. However, in practice, the spraying technique is more convenient. Furthermore, the specific influence of lighting on plants’ uptake of ZnO and CuO NPs has not been explored yet. Considering the lack of knowledge, a study was conducted to investigate how different HPS and LED lighting combined with CuO and ZnO NPs influence the elemental composition of ice plants (*Mesembryanthemum crystallinum* L., family Aizoaceae, Caryophyllales). The ice plant is a crucial research subject due to its unique ability to grow in salinity soil or water [23,24], switch from C3 photosynthesis to Crassulacean acid metabolism [23], and it contains d-pinitol, which is important for people with diabetes because it contributes to the regulation of blood sugar, and is also a source of mineral elements and antioxidants [25] to supplement the daily diet. The plant’s distribution and growth are influenced by various environmental factors, making it an essential species for ecological studies [26]. Additionally, the ice plant’s potential as a highly salt-tolerant crop and its beneficial properties for human health make it a promising candidate for future agricultural use [24].

The main research objectives were (i) to determine the possible accumulation of Zn and Cu in ice plants treated with ZnO and CuO NPs, and to find out how different lighting characteristics can lead to Zn and Cu accumulation; (ii) to investigate the possible risk assessment to human health of increased uptake of ZnO and CuO NPs in the edible parts of ice plants; and (iii) to determine the effect of ZnO and CuO NPs, by evaluating the correlation with other microelements, macroelements, and growth indicators. This study hypothesized that the accumulation of Zn and Cu in ice plant tissues can be intensified by exposure to ZnO and CuO NPs, and by adjusting lighting conditions while keeping the hazard quotients within safe limits.

## 2. Results

This research investigated the impact of lighting and NPs on the growth rates and elemental composition of ice plant leaves. The growth parameter results show that LED lighting significantly impacted the fresh shoot weight of the ice plant, leading to an increase of 12 to 80%, regardless of whether NPs were used. The highest fresh weight of the roots was observed under the influence of HPS lighting and CuO NPs, while the lowest was under the influence of ZnO NPs. From Figure 1, it is evident that the effect of NPs under different lighting conditions is noticeable. Under HPS lighting, the CuO NP-suspension had a positive impact on ice plant growth by increasing the leaf area (by 70%), FW, and DW of both shoots (by 40 and 15%, respectively) and roots (by 42 and 53%, respectively) compared to the plants that were not treated. Meanwhile, ZnO NPs did not show statistically significant differences in growth rates from unaffected-plant NPs.

Under LED lighting, the treatment of CuO NPs showed statistically significant differences by reducing the FW of the shoot part by 12% and DW of roots by 25%, and increasing leaf area by 23% compared to plants that were not treated with NPs.

Moreover, a strong positive correlation (Table 1) was found between the FW and DW of the shoot and root. Thus, the results correspond proportionally to each other in Figure 1. A strong positive correlation was also found between leaf area, shoot, and root FW.

It is noticeable that the utilization of LED lighting has led to a statistically significant increase in magnesium (Mg) by 36–50% and potassium (K) by 3–16% content in the leaves of ice plants (Table 2). On the other hand, HPS lighting combined with CuO NPs treatment increased Ca content by 39%. In addition, phosphorus (P) showed a strong positive correlation (Table 1) with K and zinc (Zn) and a weaker positive correlation with Mg. Furthermore, Ca exhibited a positive correlation with copper (Cu) and sodium (Na) elements, while showing a strong negative correlation with K, Mg, and molybdenum (Mo).

LED lighting combined with ZnO NPs application had a particularly significant effect on Zn accumulation in plant leaves, increasing it by 66% compared to the influence of HPS and ZnO NPs treatment. However, HPS lighting combined with CuO NPs were more efficient, increasing Cu content in ice plant leaves by 40% compared to the effect of LED and CuO NPs. It should be noted that a strong positive correlation (Table 1) was found between Zn, P, and B. A positive correlation was found between Cu and Ca, but a negative correlation was found with Mg.

The content of Fe and Mo increased in ice plant leaves under the influence of LED lighting and CuO NPs by 266 and 54%, respectively, compared to the HPS and CuO NPs treatment. A strong positive correlation of Fe with K, Mg, Mn, and Mo were found. Mo was strongly positively correlated with Fe, K, Mg, and Mn, while a negative correlation was found with B, Ca, and Na. Mn content increased by 49% in the leaves of ice plants under the combined effect of LED and ZnO NPs, compared to the effect of HPS and ZnO NPs. In addition, Mn positively correlated with Fe, K, Mg, and Mo (Table 3).

The results revealed that LED lighting positively affected the leaf area, the DW and FW of shoots and roots, and the amounts of K, Mg, Na, Zn, Fe, Mn, and Mo. HPS lighting positively affected the Ca and Cu content in ice plant leaves, while P and B lighting had no effect. CuO NPs significantly positively affected the leaf area, the DW and FW of shoots, the FW of roots, and the amount of Cu and Mo content (Table 4). The influence of NPs was not determined on root DW, Ca, K, Mg, Na, Fe, and Mn amounts. ZnO NPs significantly positively affected the amount of P, Zn, and B elements. Combined factors show that LED illumination without NPs (the number of values of a and ab in the interactions of the factors—10), with CuO NPs (6), with ZnO NPs (11), and HPS with CuO NPs (6) resulted in larger plants and the accumulation of more elements (Table 4).

The ability of plants to absorb Cu was strongly determined by HPS lighting and the application of CuO NPs. HQ increased from 0.032 (HPSx0NPs) and 0.002 (LEDx0NPs) to 0.068 and 0.051, respectively, when using CuO NPs, indicating that as a metal, this amount would not harm human health and does not exceed the dangerous limit of 1.

Zn accumulation was positively affected by LED illumination combined with the application of ZnO NPs in ice plant leaves. BCF increased by 72% under combined exposure to LED and ZnO NPs than under combined exposure to HPS (Table 5). HQ was found to be the highest at 0.075, when ice plants were grown under LED lighting and foliar applications of ZnO NPs, and 0.045 under combined HPS with ZnO NPs exposure, but the indices are still within safe limits. BCF_Cu_ and HQ_Cu_ found the strongest positive correlations (Table 1) with the leaf area, Cu, and Ca; for BCF_Zn_ and HQ_Zn,_ the strongest positive correlations were found with P, B, and Zn.

## 3. Discussion

### 3.1. Combined ZnO, CuO NPs, and Lighting Impact on Cu and Zn Accumulation in Plants

There are articles about the influence of different lighting on ice plants’ growth indices and metabolites. For example, a higher ratio of R to B (70–90% to 10–30%, respectively) lighting at 150 μmol m^−2^ s^−1^ intensity with a 14 h photoperiod [25] led to the higher productivity and better quality of ice plants. A similar ratio of R to B (90%:10%) positively influenced the results of photosynthesis and fresh and dry mass, which were determined when growing ice plants in aeroponics at 350 m^−2^ s^−1^ intensity. However, there are considerable gaps in knowledge about the influence of lighting on the accumulation of minerals in ice plants. Thus, this study expands the knowledge about LED lighting with a ratio of white (65%), blue (5%), and red (30%) having a positive influence on all elements except P, Cu, and B in ice plant leaves (Table 4) compared with HPS lighting. Other research showed that a higher percentage of blue light in the illumination spectrum resulted in a higher content of minerals in various plants [27,28]. The literature data proposed that blue light, through the control of the blue light receptor phototropin (Phot 1 and Phot 2) affects stomatal opening and membrane transport activity and promotes the flux of ion transport [29,30,31,32].

On the other hand, HPS lighting positively affected Ca content. Such a trend was established for red pak choi microgreens cultivated indoors [33]. It is known that HPS lighting had a low red/far red ratio, a low blue-light emission, and a high percentage in the yellow and red parts of the spectrum. Positive red-light effects were determined on Ca, K, Mg, and P contents, which were higher in microgreens [34]. 

In our experiments, using 800 ppm ZnO NPs increased Zn content by 81% under HPS illumination and 356% when plants were grown under LED. Other researchers [21] investigated black mustard plants grown in soil under natural light with ZnO NPs concentrations of 200, 400, and 600 mg kg^−1^ accumulated 32, 98, and 277% more Zn in the edible parts of the plants, respectively. Cucumbers grown in greenhouse soil under the natural light of 300 μmol m^−2^ s^−1^ (recalculated as a photoperiod of 16 h at 17 DLI) sprayed with ZnO NPs at 400 and 800 mg kg^−1^ increased Zn content by 6.3 and 9 times in their leaves [15]. As we can see, our applied LED lighting can be compared with natural daylight to accumulate more Zn in plants. In another study [14], plants grown under a 16 h photoperiod using fluorescent lamps (18 W) with a light intensity of 40.5–54 μmol m^−2^ s^−1^ at a maximum concentration of 900 mg kg^−1^ increased the Zn content in radish by 11 times; tomato by 99 times; corn, lettuce, and beets by 7 times; wheat by 10 times; and beans, peas, and cucumber by 1.5 times. Furthermore, plants grown under fluorescent light at an intensity of 250 μmol m^−2^ s^−1^ and a photoperiod of 16 h were applied with 100 ppm ZnO NPs. It was found that the Zn content increased in cilantro by 1080%, parsley by 300%, and spinach by over 33% [17] compared with non-treated plants. Significant differences prevail between plant species due to their morphological characteristics and cultivation differences.

The HPS lighting showed a more significant positive effect on Cu accumulation when the plants were sprayed with a concentration of 30 ppm CuO NPs compared to LED lighting. For example, plants grown with LED lighting accumulated 0.5–1.4 µg g^−1^ of Cu, while under HPS lighting, NPs-untreated ice plants accumulated 10.1 µg g^−1^ of Cu in leaves. Compared with the works of other scientists [21], it can be noted that concentrations of 12.5, 25, and 50 mg kg^−1^ of CuO NPs increased the amount of Cu in black mustard by 33, 48, and 67%, respectively. In our experiment, using HPS lighting and a CuO NPs concentration of 30 ppm, Cu accumulated 100% more than untreated plants. Lettuce accumulated [20] about 15 and 20% more Cu in leaves than untreated plants when grown at a light intensity of 200 μmol m^−2^ s^−1^, a photoperiod of 14 h, and treated with 200 and 400 mg kg^−1^ CuO NPs through the soil. Other researchers [18] have investigated that using a concentration of 100 mg kg^−1^ CuO NPs on burr medica in a controlled climate chamber with a 16 h photoperiod and nearly 500 μmol m^−2^ s^−1^ increased the Cu content in leaves by up to 180%. As seen here, most of the experiments’ descriptions do not indicate how many and what wavelengths were used in the illuminations, making it very difficult to compare the results. However, considering that in our study, a relatively low concentration of CuO NPs was used and the fact that the Hoagland solution itself also contained a Cu concentration of about 0.08 ppm, the uptake of which could be affected by illumination, compared to other studies, the increase in Cu content in the ice plant is quite competitive. Overall, the effects of nanoparticles on Zn and Cu accumulation in plants are determined by plant species, environmental factors such as nanoparticle concentrations and properties, lighting, soil pH, and treatment methods.

### 3.2. ZnO and CuO NPs Impact on Other Micro- and Macroelements in Plants 

It should be emphasized that the spraying of ice plants with ZnO NPs increased the absorption of P and Zn and decreased Mo compared to the control (Table 3). CuO NPs affected the elemental composition of ice plants by increasing Cu and decreasing B elements. A positive correlation was assessed between the Cu bioaccumulation factor and Ca, and a negative correlation for the Mg element. ZnO NPs had a positive correlation with P and B elements. Other researchers [16] found that ZnO NPs increased the amount of Zn, N, and K, but decreased the amount of P in sorghum plant shoots. In addition, ZnO NPs increased plants’ Mg, B, Cu, and Mn content. Although another study [18] conducted with burr medic highlights that ZnO NPs positively affected the accumulation of P, K, Mg, Ca, Fe, Ni, Zn, Cu, and Mn, the effect depended on the concentration used. CuO NPs positively [18] influenced the accumulation of P, K, Mg, Ca, Zn, Cu, and Mn. In general, our experiments prove that lighting influenced the accumulation of elements other than Zn and Cu more than NPs.

### 3.3. ZnO and CuO NPs’ Impact on Calculation-Based Risk Assessment to Human Health

Our research agrees with others because the hazard quotient (HQ) under different ZnO NPs and CuO NPs treatments were less than 1. Our calculated maximum HQ values were 0.075 and 0.068 using an 800 ppm concentration of ZnO NPs and 30 ppm CuO NPs, where ice plants accumulated the highest amounts of Zn and Cu under optimal conditions. Others [18] estimated that using a concentration of 25–200 mg/L ZnO NPs could raise the HQ from 0.019 to 0.052, while 10–100 mg/L CuO NPs affected the HQ from 0.02 to 0.035. Spraying the sorghum [16] plant with the 100 mL suspension containing 48 mg of ZnO NPs HQ was calculated to be about 0.016 in leaves and 0.042 in grains. The calculated HQ in cucumber fruits were 0.031 and 0.034 when grown in soil with 400 and 800 mg/L ZnO NPs, respectively [15]. These results indicated no risk of overdosing on Zn and Cu elements in the human body exposed to the ZnO and CuO NPs-treated plants. A more significant concern is whether and how NPs are affected inside the plant under different conditions than when in water and in contact with plant metabolites. 

The most critical processes in NPs are dissolution and sulfidation, which determine the surface properties, toxicity, and durability of NPs. This is especially important for NMs made from B-class soft metal cations such as Ag, Zn, and Cu, as they form partially soluble metal oxides and have a strong affinity for inorganic and organic sulfide ligands [35]. The dissolution and release of cations usually express the NM toxicity of class B metals. Complete dissolution may allow their effects to be predicted using existing metal speciation and exposure models. However, the affinity of B-class metals for electron-dense sulfur molecules makes them highly reactive with sulfur-containing biomacromolecules (phytochelatines) [36] and inorganic sulfur in sediments, soils, and air [35,37]. The coat of NPs, which maintains its shape and size, can be affected by lighting, pH, and temperature, which can influence oxidative processes (release ions) and affect reactions with ligands of various substances, promoting or inhibiting oxidation and dissolution. For example, ZnO NPs of the same size as those used in our studies (35–45 nm size) at a concentration of 100 ppm showed [38] the best solubility of 91% per 24 h in a phagolysosomal model fluid (pH 4.5); 19% per 24 h in cell culture medium (pH 7.4) due to the amino acids and proteins they contain; the solubility in Gamble’s liquid (pH 7) was 4.6% per 24 h; and the lowest solubility in ultrapure water (pH 7.8) was 3% per 24 h. Given this dependence, the pH of lettuce is 5.97, and the pH of basil is 6.08 [39]. However, each plant organelle has a different pH, which depends on the watering regime, fertilization, temperature, lighting, and stress factors [40]. CuO NPs (30 ppm) were most dissolved into ions when the pH ranged from 3 to 6 [7], and then the released Cu^2+^ was transformed into soluble hydroxide, hydroxy complex, and Cu^2+^ in the supernatant. Cu^2+^ ion concentrations ranged from 1.11 to 31.9 and 1.35 to 38.3 μg L^−1^ after 2 and 10 days of incubation, respectively [41]. In addition, studies [41] have shown that the original free Cu^2+^ ion formed complexes with natural organic matter or anions in reclaimed waters. Notably, the solubility of NPs strongly depends on the size [42] and the pH [43] of the medium into which they enter and the substances contained in them. 

No other studies were found combining lighting and NPs influence on HQ. Although our study proves that LED lighting affected the higher accumulation of Zn in ice plants, HPS lighting had a more significant effect on Cu, and lighting itself did not cause an excess of HQ that would be harmful to human health. Considering this, our study significantly contributes to the scientific directions of precision horticulture, safe food, and nanotechnology application.

## 4. Materials and Methods

### 4.1. Nanoparticles Preparation and Characteristics

The commercial zinc (ZnO) and copper (CuO) oxide nanoparticles (NPs) used for plant exposure in this study were purchased from US Research Nanomaterials (Inc., Houston, TX, USA). Suspensions of ZnO (size: 35–45 nm, 99% purity)—800 ppm and CuO (size: 40 nm, 99% purity)—30 ppm NPs were prepared in deionized water. The suspensions were prepared in 300 mL flasks by weighing ZnO and CuO NP powders, respectively, which were weighed using a highly sensitive balance (Radwag AS 220 R2 PLUS, RADWAG Balances and Scales, Torunska, Poland) and an antistatic ionizer (DJ-04 Antistatic Ionizer, RADWAG Balances and Scales, Torunska, Poland) to remove the static charge of the powder particles. The flasks with suspensions were placed in an ultrasonic bath (Sonerex super ultrasonic bath 80 W, Weidinger GmbH, Gernlinden, Germany) and suspended for 60 min. Immediately afterward, the NPs’ size and suspension stability were measured using a Delsa™Nano Submicron Particle Size (Beckman Coulter Instruments Corporation, Fullerton, CA, USA) and a Zeta Potential device (Dispersion Technology Inc., Bedford Hills, New York, NY, USA). Table 6 shows the positive particle surface charge of the ZnO and CuO-NP suspensions. The suspensions were stable, according to the zeta potential. In addition, the polydispersity index (PDI) showed that NP suspensions were monodisperse.

Plants were sprayed immediately after the ultrasonic bath using automatic sprayers (Rechargeable electric sprayer,1 L, 3.6 V, Nozzle hole diameter: 13 mm, Yato, Haiyan, Jiaxing, China) to the full surface maturity during the first half of the day. Before spraying, the plant systems were covered with a plastic sheet to protect the hydroponic solution and roots from exposure to NPs. It should be noted that certain individuals who carried out the spraying followed all safety requirements, wearing a full protective suit, gloves, and a respirator.

### 4.2. Plant Growth Conditions, Lighting, and Nanoparticle Treatments

Experiments were conducted at the Institute of Horticulture in the Lithuanian Research Centre for Agriculture and Forestry. The study was performed in a controlled environment plant growth chamber measuring 4 m by 6 m with a height of 3.2 m. Seeds of the ice plant (*Mesembryanthemum crystallinum* L.) were obtained from CN Seeds, Ely, UK. The 200 rockwool cubes measuring 2.5 cm × 2.5 cm × 3.0 cm were used as the growing medium. Before use, the rockwool cubes were soaked in deionized water with an adjusted pH of 5.0 by adding sulfuric acid and placed in a plastic tray. Germinated seedlings were grown in this setup for 29 days at a temperature of 25 ± 1 °C and humidity of 60 ± 5%. The germinated plants were watered with enough hydroponic solution to cover one centimeter of the bottom of the rockwool cubes daily. Then, they were transferred to Ebb-type hydroponic systems with 80 L containers containing a hydroponic solution made of deionized water and nutrients in the following quantities (mg L^−1^): N (120), Ca (88), P (20), K (128), Mg (40), S (53), B (0.16), Mo (0.2), Mn (0.08), Cu (0.08), Fe (1.6), and Zn (0.8). The electrical conductivity (EC) of the nutrient solution was 1.4 mS cm^−1^, and the pH was measured daily using a portable meter (GroLine HI9814, Hanna Instruments, Woonsocket, RI, USA) and adjusted to 6.0 using sulfuric acid or sodium bicarbonate. During the experiment, the plants were exposed to two different lighting conditions: a combination of white—380–760 nm (4000 K), blue—455 nm, and red—660 nm light-emitting diodes (LEDs, OSRAM Oslon SSL, Ecolight, Vilnius, Lithuania) at the ratio of 13:1:6, respectively, and high-pressure sodium lamps—2050 K (HPS, SON-T Agro, 400 W, Philips, Eindhoven, The Netherlands) at a photosynthetic photon flux density (PPFD) of 250 ± 5 μmol m^−2^ s^−1^ with a 16 h photoperiod. PPFD and spectra (Figure 2) were measured at the plant growth level using a portable spectroradiometer (WaveGo, Wave Illumination, Oxford, Oxfordshire, UK). After eight days of ice plant cultivation in Ebb hydroponic systems, the plants were treated with ZnO and CuO-NP suspensions (preparation described in Section 4.1) by spraying and were allowed to grow for five more days until the end of the experiment. Then, the plant growth parameters were measured, and raw materials for the analysis were collected.

### 4.3. Growth Characteristics

The growth characteristics of the ice plant were studied by measuring various parameters such as the number of leaves, fresh (FW) and dry weights (DW) of the shoots and roots, and leaf area. The study involved 10 adult plants per treatment. Adult plants were described as side shoots with secondary leaves but no flowers and with primary leaves [44]. The FW was measured using an electronic scale (Mettler Toledo, ML104T/00; Mettler-Toledo, Columbus, OH, USA). The DW was determined by drying the divided samples of shoots and roots for 48 h in a drying oven (Venticell-222, Medcenter Einrichtungen, Gräfeling, Germany) at 70 °C. The leaf area was measured using a leaf area meter (CI-202 Laser Area Meter; CID BioScience, Camas, WA, USA). 

### 4.4. Elemental Composition of Ice Plant

The macro-and microelement quantities in ice plant leaves were determined using microwave digestion combined with inductively coupled plasma optical emission spectrometry. The shoots of the ice plants were harvested, gently rinsed with ultrapure water, and dried at 70 °C for 48 h, then ground to powder using a centrifugal mill with a ZM 200 rotor at 15,500 rpm (Ultra Centrifugal Mill ZM 300, RETSCH GmbH, Haan, Germany). Complete digestion of dry ice plant material (0.3 g) was achieved with 8 mL of 65% HNO_3_ using the microwave digestion system Multiwave GO (Anton Paar GmbH, Graz, Austria). The digestion program was as follows: (1) 170 °C reached within 3 min, digested for 10 min; and (2) 180 °C reached within 10 min, digested for 10 min. Full-digested samples were diluted to 50 mL with deionized water. The elemental profile was analyzed by an ICP–OES spectrometer (Spectro Genesis, SPECTRO Analytical Instruments, Kleve, Germany). The operating conditions employed for ICP-OES determination were 1300 W RF power, 12 L min^−1^ plasma flow, 1 L min^−1^ auxiliary flow, 0.8 L min^−1^ nebulizer flow, and 1 mL min^−1^ sample uptake rate. The analytical wavelengths chosen were 213.618 nm for P, 766.491 nm for K, 279.079 nm for Mg, 589,592 nm for Na, 445.478 nm for Ca, 324.754 nm for Cu, 257.611 nm for Mn, 259.941 nm for Fe, 213.856 nm for Zn, 249.773 nm for B, and 208,414 nm for Mo. The operating conditions employed for the ICP-OES were as follows: 1.3 kW RF power, 1.0 L min^−1^ auxiliary argon (Ar) flow, 0.80 L min^−1^ nebulizer Ar flow, 12 L min^−1^ coolant Ar flow, and axial plasma configuration. Each sample was analyzed in triplicate. The calibration standards were prepared by diluting a stock multi-elemental standard solution (1000 mg L^−1^) in 6.5% (*v*/*v*) nitric acid and by diluting stock phosphorus and standard sulfur solutions (1000 mg L^−1^) in deionized water. The calibration curves for all the studied elements ranged from 0.01 to 400 mg L^−1^. The contents of macro and microelements in the DW of ice plants are presented [45,46].

### 4.5. Bio-Concentration Factor and Intake Risk Assessment

Bioaccumulation is the ability of plants to absorb elements and retain them. The efficiency of this process depends on environmental conditions, and the plant type has the most influence on the elements’ retention. Due to the complexity of the methods, the bioconcentration factor (BCF) of specific elements was calculated to assess the environmental risk that may arise from the substances under investigation. Depending on the component, a high BCF value means a low element solubility in water and a high relative octanol–water partition coefficient besides a high soil adsorption coefficient.

Therefore, BCF was calculated as an index of the ice plants’ ability to accumulate zinc (Zn) and copper (Cu). It was calculated as the ratio of the Zn or Cu concentration (mg L^−1^) in the hydroponic solution and the Zn or Cu concentrations (mg kg^−1^) in the ice plant [47]:(1)$$\mathrm{BCF}=\frac{C_{shoot}}{C_{soil}}$$

The average daily intake (mg kg^−1^ day^−1^) of potentially toxic metals by consuming leaves of ice plant after foliar application of ZnO NPs was calculated by the following equation [48]:(2)$$ADI=\frac{C_{m}\times C_{f}\times IR}{Bw}$$

C_m_—the metal concentration in a plant (mg kg^−1^) on a dry weight basis. 

C_f_—the conversion factor (0.085) to convert the fresh to dry weight. 

IR—the ingestion rate of vegetables. 

Bw—the average body weight for an adult is 70 kg. 

The average daily intake of leafy vegetables was estimated to be 100 g (0.1 kg person^−1^ day^−1^).

The risk of non-carcinogenic health effects is often evaluated from hazard quotients (HQ), which are the ratio of the daily intake (often the average daily intake, ADI) to a toxicological reference dose (RfD) according to the following equation [48]:(3)$$HQ=\frac{ADI}{RfD}$$

RfD—the oral reference dose for Zn is 0.3 mg kg^−1^ day^−1^ [49] and Cu 0.04 mg kg^−1^ day^−1^ [50].

If the value of HQ is less than 1, it is assumed to be safe from the risk of non-carcinogenic effects. Conversely, if the HQ is equal to or higher than 1, it indicates a potential risk for some exposed individuals to experience adverse health effects.

### 4.6. Statistical Analysis

MS Excel Version 2010 and XLStat 2020 Data Analysis and Statistical Solution for Microsoft Excel (Addinsoft, Paris, France) statistical software were used for data processing. Analysis of variance (ANOVA) was carried out along with the Tukey multiple comparison test for statistical analyses (*p* ≤ 0.05).

## 5. Conclusions

In general, our experiments prove that the accumulation of elements other than Zn and Cu was more influenced by lighting than treatment with NPs. LED lighting combined with the application of ZnO NPs had a particularly significant effect on Zn accumulation in plant leaves, and Cu accumulation was higher when plants were treated with the combined effect of CuO NPs and HPS illumination. The calculated hazard quotients showed that the limits are not exceeded when applying our selected concentrations and growth conditions on ice plants. Ice plants had a more significant positive effect on the accumulation of macro- and microelements under LED lighting than HPS. CuO and ZnO NPs had the most potent effect on increasing their respective microelements.

## Figures and Tables

**Figure 1 plants-13-00681-f001:**
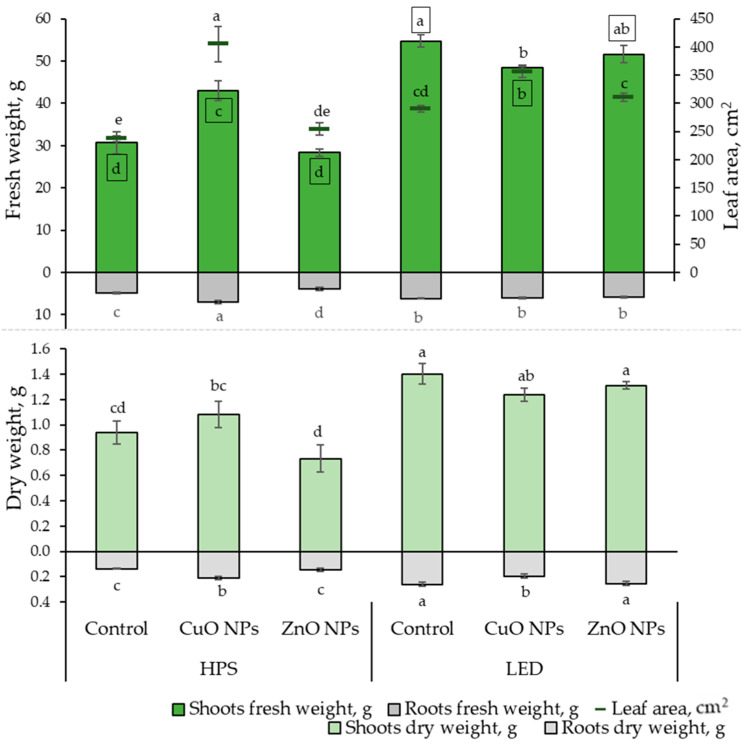
Effects of different lighting (HPS and LED) and nanoparticles (30 ppm CuO and 800 ppm ZnO) on ice plant (*n* = 10) growth characteristics: leaf fresh and dry weight, root fresh and dry weight, and leaf area. The data were processed using analysis of variance (ANOVA) and the Tukey (HSD) multiple range test at the confidence level *p* = 0.05. Different letters represent significant differences. The statistical reliabilities presented in squares are assigned to fresh weight. NPs—nanoparticles, control—plants treated with deionized water, HPS—high-pressure sodium lamps, LED—light-emitting diodes.

**Figure 2 plants-13-00681-f002:**
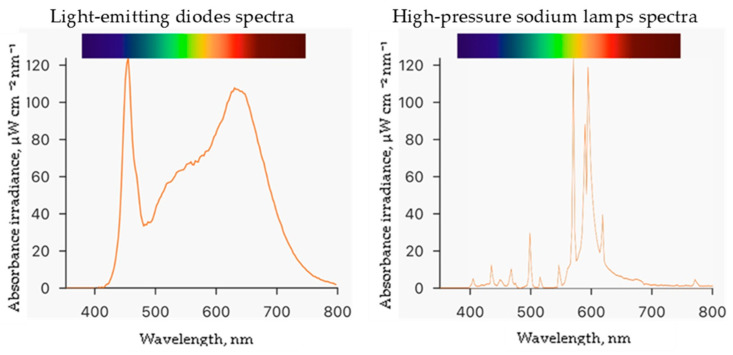
Spectra of different lighting light-emitting diodes and high-pressure sodium lamps.

**Table 1 plants-13-00681-t001:** The correlation matrix heatmap displays the Pearson correlation coefficient values for all parameters studied. Positive values are shown in red, while negative values are displayed in blue. The correlation coefficient ranges are from −1 to 1. A perfect negative linear correlation between variables indicates a value of −1, while 1 indicates a perfect positive linear correlation between variables (Color meanings are represented in the scale). A value of 0 indicates no relationship between the studied variables. Level of significance * *p* < 0.05, ** *p* < 0.01, *** *p* < 0.001.

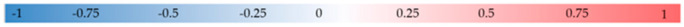
	Shoots FW	Shoots DW	Roots FW	Roots DW	Leaf Area	BCF CuO	HQ CuO	BCF ZnO	HQ ZnO	P	B	Ca	Cu	Fe	K	Mg	Mn	Na	Zn		Mo
Shoot FW	-	***	***	***	**									*	**	***	***				***
Shoot DW	-	***	***										*	**	***	***				***
Roots FW	-	***	***																
Roots DW	-	*											***	*				
Leaf area	-	**	**				**		**								
BCF CuO	-	***					**	***			**					
HQ CuO	-					**	***			**					
BCF ZnO	-	***	***	***								***		
HQ ZnO	-	***	***								***		
									P	-					***	*			***		
										B	-								***		*
											Ca	-	**		***	**		***			**
												Cu	-			**					
													Fe	-	***	***	**				***
														K	-	***	**				***
															Mg	-	***	**			***
																Mn	-				**
																	Na	-			**
																		Zn	-		
																			Mo		1

DW represents dry weight, FW represents fresh weight, BCF—bioconcentration factor, HQ—hazard quotients, P—phosphorus, B—boron, Ca—calcium, Cu—copper, Fe—iron, K—potassium, Mg—magnesium, Mn—manganese, Na—sodium, Zn—zinc, and Mo—molybdenum.

**Table 2 plants-13-00681-t002:** Effects of different lighting (HPS and LED) and nanoparticles (30 ppm CuO and 800 ppm ZnO) on ice plant macroelements, mg g^−1^ DW. Different letters represent significant differences. The data were processed using analysis of variance (ANOVA) according to the Tukey (HSD) multiple range test at the confidence level of *p* = 0.05.

Lighting	NPs	P	Ca	K	Mg	Na
HPS	0	0.492 a	6.181 ab	52.912 ab	3.344 b	5.348 ab
	CuO	0.478 a	8.365 a	51.140 b	3.045 b	5.543 a
	ZnO	0.557 a	7.048 ab	54.941 ab	3.149 b	5.514 ab
LEDs	0	0.512 a	5.769 b	59.610 a	4.550 a	5.333 ab
	CuO	0.532 a	5.997 b	59.328 a	4.500 a	5.130 b
	ZnO	0.576 a	6.693 ab	56.539 ab	4.743 a	5.323 ab

HPS—high-pressure sodium lamps, LEDs—light-emitting diodes, NPs—nanoparticles, 0—plants treated with deionized water, DW—dry weight, P—phosphorus, Ca—calcium, K—potassium, Mg—magnesium, and Na—sodium.

**Table 3 plants-13-00681-t003:** Effects of different lighting (HPS and LED) and nanoparticles (30 ppm CuO and 800 ppm ZnO) on ice plant microelements, µg g^−1^ DW. Different letters represent significant differences. The data were processed using analysis of variance (ANOVA) according to the Tukey (HSD) multiple range test at the confidence level of *p* = 0.05.

Lighting	NPs	Zn	Cu	Fe	Mn	B	Mo
HPS	0	82.015 c	10.645 c	40.654 b	49.410 ab	11.923 a	12.600 c
	CuO	80.663 c	22.355 a	40.820 b	43.861 ab	4.811 b	11.426 c
	ZnO	147.593 b	11.145 c	48.972 ab	38.802 b	11.038 a	11.077 c
LEDs	0	53.981 c	0.532 d	73.900 ab	50.285 ab	6.044 ab	16.314 ab
	CuO	66.544 c	16.897 b	149.419 a	50.862 ab	3.120 b	17.568 a
	ZnO	245.847 a	1.384 d	99.997 ab	57.738 a	11.752 a	13.847 bc

HPS—high-pressure sodium lamps, LEDs—light-emitting diodes, NPs—nanoparticles, 0—plants treated with deionized water, DW—dry weight, Zn—zinc, Cu—copper, Fe—iron, Mn—manganese, B—boron, and Mo—molybdenum.

**Table 4 plants-13-00681-t004:** Summary of all pairwise comparisons for individual factors and their interactions. Different letters represent a reliable difference between variants; descending letters of the alphabet correspond to descending numerical values. Bold letters indicate statistically significant differences between compared variants.

Lighting	HPS	LED	0 NPs	CuO NPs	ZnO NPs	HPS × 0 NPs	HPS × CuO NPs	HPS × ZnO NPs	LED × 0 NPs	LED × CuO NPs	LED × ZnO NPs
Leaf area	b	**a**	b	**a**	b	**e**	**a**	ed	cd	**b**	**c**
Shoots FW	b	**a**	b	**a**	b	**d**	**c**	**d**	**a**	**b**	ab
Shoots DW	b	**a**	**a**	**a**	b	cd	bc	**d**	**a**	ab	**a**
Roots FW	b	**a**	b	**a**	c	**c**	**a**	**d**	**b**	**b**	**b**
Roots DW	b	**a**	**-**	**-**	**-**	**c**	**b**	**c**	**a**	**b**	**a**
P	**-**	**-**	b	b	**a**	**-**	**-**	**-**	**-**	**-**	**-**
Ca	**a**	b	**-**	**-**	**-**	ab	**a**	ab	**b**	**b**	ab
K	b	**a**	**-**	**-**	**-**	ab	**b**	ab	**a**	**a**	ab
Mg	b	**a**	**-**	**-**	**-**	**b**	**b**	**b**	**a**	**a**	**a**
Na	b	**a**	**-**	**-**	**-**	ab	**a**	ab	ab	**b**	ab
Zn	b	**a**	b	b	**a**	**c**	**c**	**b**	**c**	**c**	**a**
Cu	**a**	b	b	**a**	b	**c**	**a**	**c**	**d**	**b**	**d**
Fe	b	**a**	**-**	**-**	**-**	**b**	**b**	ab	ab	**a**	ab
Mn	b	**a**	**-**	**-**	**-**	ab	ab	**b**	ab	ab	**a**
B	**-**	**-**	**a**	b	**a**	**a**	**b**	**a**	ab	**b**	**a**
Mo	b	**a**	**a**	**a**	b	c	c	c	ab	**a**	bc

FW represents fresh weight, DW represents dry weight, P—phosphorus, B—boron, Ca—calcium, Cu—copper, Fe—iron, K—potassium, Mg—magnesium, Mn—manganese, Na—sodium, Zn—zinc, and Mo—molybdenum. Different colors identify variants between which statistical analysis was performed.

**Table 5 plants-13-00681-t005:** Effects of different lighting (HPS and LED) and nanoparticles (30 ppm CuO and 800 ppm ZnO) on bioconcentration factor (BCF), average daily intake (ADI), and the risk to humans’ health evaluated from hazard quotients (HQ). Different letters represent significant differences. The data were processed using analysis of variance (ANOVA) according to the Tukey (HSD) multiple range test at the confidence level of *p* = 0.05.

Lighting	NPs	ADI_Cu_	HQ_Cu_	BCF_Cu_	ADI_Zn_	HQ_Zn_	BCF_Zn_
HPS	0	0.001 c	0.032 c	0.13 c	0.010 c	0.025 c	0.10 c
	CuO	0.003 a	0.068 a	0.28 a	0.010 c	0.025 c	0.10 c
	ZnO	0.001 c	0.034 c	0.14 c	0.018 b	0.045 b	0.18 b
LEDs	0	0.001 d	0.002 d	0.01 d	0.007 c	0.016 c	0.07 c
	CuO	0.002 b	0.051 b	0.21 b	0.008 c	0.020 c	0.08 c
	ZnO	0.001 d	0.004 d	0.02 d	0.030 a	0.075 a	0.31 a

HPS—high-pressure sodium lamps, LEDs—light-emitting diodes, NPs—nanoparticles, and 0—plants treated with deionized water.

**Table 6 plants-13-00681-t006:** Properties of ZnO and CuO NPs suspensions in deionized water: polydispersity index (PDI) and zeta potential (ZP) results represent the mean ± standard error and percentage of nanoparticles between 1 and 100 nm in the suspension.

	ZnO, 35–45 nm, 800 ppm	CuO, 40 nm, 30 ppm
PDI	0.293 ± 0.061	0.267 ± 0.029
ZP (mV)	32.61 ± 1.002	10.48 ± 1.916
Particle size in suspension up to 100 nm; %	58.5	68.2

## Data Availability

All the data are available upon request.

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
