# Peer review of "Nanoparticle Effects on Ice Plant Mineral Accumulation under Different Lighting Conditions and Assessment of Hazard Quotients for Human Health"

_plants, 2024, doi:10.3390/plants13050681_

Round 1

Reviewer 1 Report

Comments and Suggestions for Authors

The research aimed to assess the response of ice plants to cultivation in various lighting conditions and fertilization with zinc and copper nanoparticles. Laboratory tests were performed correctly. Appropriate statistical methods were used to develop the research results. The research results and discussion were properly presented. The conclusions refer to the research results.

An interesting article, especially since the use of nanoparticles in the cultivation and protection of plants, including vegetables, is becoming more and more popular. The only thing missing is a research hypothesis, which I propose to add in line 97.

With minor corrections, the work may be published in the Plants journal

Comments on the Quality of English Language

English is understandable.

Author Response

Thank you for your valuable comments and appreciation of our work. The manuscript was supplemented with a hypothesis in line 97: "This study hypothesized that the accumulation of Zn and Cu in ice plant tissues can be intensified by exposure to ZnO and CuO NPs and by adjusting lighting conditions while keeping the hazard quotients within safe limits."

Reviewer 2 Report

Comments and Suggestions for Authors

This study investigated the effects of different HPS and LED lighting combined with CuO and ZnO NPs on elemental composition of ice plants. The article submitted for review is very important from the point view of studying the effect of foliar-supplied NPs on heavy metal accumulation in plants as the uptake of elements by plants can be associated with the availability of other elements.

I strongly suggest authors to introduce more keywords. The usefulness of keywords is to make the article both more and more easily searchable visible after its publication through commonly used search engines.

The introduction is interesting, but in my opinion, it does not fully cover the topic. Moreover, out of cited items, some are older than 5 years. The authors refer to some very old literature (item 12 & 23). Can those items be replaced with newer one?

It should be noted that the authors described Materials and Methods very thoroughly, as well as the description and graphic representation of the results were presented very well. I am asking for a deeper description, taking into account my suggestions above, with post new items.

Some of the recent literature can be cited to discuss the uptake, portioning and how they contribute to assimilates and growth of the plants.

For ex:

https://doi.org/10.24326/asphc.2021.5.4.

https://doi.org/10.3390/agronomy11020224.

Comments on the Quality of English Language

Minor editing of English language required

Author Response

Thank you for your valuable comments. The manuscript keywords were supplemented with LED, HPS, NPs, hazard quotients, Cu accumulation, and Zn accumulation.
Also, the recommendation to change several cited sources to new ones was taken into account, and the changes can be seen in lines 515 and 545: sources 12 and 23 were changed to 2023 publications.

Upon careful consideration, we have determined that the suggested articles do not align closely with the thematic focus of our publication because it is about the accumulation of elements in the fruit trees of the loquat (Eriobotrya japonica). Considering that the mineral metabolism of the ice plant edible succulent used in our study significantly differs from the accumulation of elements carried out by trees. Furthermore, your recommended publications do not mention the use of nanoparticles. We will keep these articles you recommend in mind when writing other, more related manuscripts.

Thank you once again for your time and consideration.